# Radar Multiple Bin Selection for Breathing and Heart Rate Monitoring in Acute Stroke Patients in a Clinical Setting

**DOI:** 10.3390/s26010251

**Published:** 2025-12-31

**Authors:** Benedek Szmola, Lars Hornig, Jan Paul Vox, Thomas Liman, Andreas Radeloff, Birger Kollmeier, Karen Insa Wolf, Karsten Witt

**Affiliations:** 1Department of Neurology, School of Medicine and Health Science, Carl von Ossietzky Universität Oldenburg, 26129 Oldenburg, Germany; thomas.liman@uni-oldenburg.de (T.L.); karsten.witt@uni-oldenburg.de (K.W.); 2Medizinische Physik, Carl von Ossietzky Universität Oldenburg, 26129 Oldenburg, Germany; birger.kollmeier@uni-oldenburg.de; 3Fraunhofer Institute for Digital Media Technology IDMT, Oldenburg Branch for Hearing, Speech and Audio Technology HSA, Marie-Curie-Straße 2, 26129 Oldenburg, Germany; lars.hornig@idmt.fraunhofer.de (L.H.); jan.paul.vox@idmt.fraunhofer.de (J.P.V.); karen.insa.wolf@idmt.fraunhofer.de (K.I.W.); 4Department of Neurology, Evangelical Hospital Oldenburg, Carl von Ossietzky Universität Oldenburg, 26122 Oldenburg, Germany; 5Division of Otolaryngology, Head and Neck Surgery, Carl von Ossietzky Universität Oldenburg, 26129 Oldenburg, Germany; andreas.radeloff@uni-oldenburg.de; 6Research Center of Neurosensory Sciences, Carl von Ossietzky Universität Oldenburg, 26129 Oldenburg, Germany

**Keywords:** FMCW radar, clinical monitoring, stroke, acute ischemic stroke, sleep monitoring, breathing rate, heart rate, multiple range bin selection, single range bin selection

## Abstract

Simple to use and accurate monitoring of stroke patients’ breathing and heart rate in sleep is needed to lower the risk of secondary strokes and prevent worse functional outcomes due to disturbed sleep. In this study, we computed breathing and heart rates from clinical radar sleep recordings of 49 acute ischemic stroke patients. Parallel polysomnography served as the reference for evaluation. We compared radar rates computed using previously developed multiple and single range bin selection methods. Multiple selection yielded lower mean absolute errors (breathing: 0.39 vs. 0.87 breaths per min; heart rate: 0.84 vs. 3.99 beats per min) and higher correlations with the reference (breathing: 0.95 vs. 0.85; heart rate: 0.96 vs. 0.56). However, single range bin selection produced rates for a larger proportion of recording time (breathing: 93.49% vs. 73.38%; heart rate: 81.85% vs. 19.93%). Our results indicate that multiple range bin selection provides more accurate estimates of breathing and heart rate, but it has lower temporal coverage. Easy to use radar systems could facilitate the clinical adoption of contactless breathing and heart rate monitoring in sleep, improving the care provided to stroke patients.

## 1. Introduction

Sleep disturbances and stroke risks are interlinked [1]. Although this link is recognized [2,3], sleep disturbances are often undiagnosed [4], leading to a gap in the care received by stroke patients. New simple to use approaches could spread the clinical use of sleep monitoring. By detecting sleep disturbances early on, better outcomes could be achieved for stroke patients.

A recent analysis of the global burden of diseases has found that stroke is among the leading causes of death and disability [5]. Patients with sleep disturbances have a higher chance of suffering a stroke [6]. Furthermore, patients who have suffered a stroke and are affected by sleep disturbances have worse functional outcomes and a higher risk of a secondary stroke [7]. Studies have demonstrated this link for various types of sleep disturbances [8], among them insomnia [9,10] and sleep disordered breathing, including sleep apnea and hypopnea [11,12,13]. Therefore, for patients with an elevated risk of stroke and those who have already suffered a stroke, sleep monitoring should be considered [14].

The current gold standard for sleep monitoring is polysomnography (PSG) [15,16]. PSG applies a collection of mainly contact-based sensors to monitor a wide range of physiological signals during sleep in a dedicated sleep laboratory with staff supervision. Although PSG is a sensitive method, several drawbacks limit its application: accessibility [17], patient comfort [18], and the first night effect [19,20]. Thus, alternative sleep monitoring methods are needed.

Various alternative sleep monitoring solutions have been explored in the literature. There are approaches that still use contact-based sensors but in configurations that are less uncomfortable for the patient [21]. Multiple contactless approaches have also been tested [22], such as using a camera [23], ambient microphones [24], or radars [25].

Biomedical radars have received increasing attention in the field of sleep monitoring [26,27,28,29]. They provide a contactless monitoring method that is convenient for both the patient and the caregivers. Radars are advantageous in terms of privacy concerns, as the types of radar applied for sleep monitoring do not have the angular resolution to resolve the facial features of the person. Furthermore, radars emit their own signal and, therefore, do not require additional lighting, which simplifies the recording. Different types of radars have been implemented for sleep monitoring, but frequency modulated continuous wave radars (FMCW) [30] provide a good balance between cost, complexity, and the information provided. FMCW radars transmit electromagnetic waves with a frequency that changes over a period of time. The transmitted waves and the received reflections are used to separate the space in front of the FMCW radar into multiple sections and to detect objects in these sections. These sections are referred to as range bins.

In a previous publication [31], we introduced multiple range bin selection and tested its feasibility in a small group of healthy volunteers sleeping in an office setting. We developed multiple range bin selection to make breathing and heart rate computation more robust by using all valid signals reflected from the person’s body. In contrast, with the more common single range bin selection method [25,29,32], only one range bin’s signal is used. When the radar is set up at the foot end of the bed and pointed at the person at an oblique angle, the FMCW radar’s range bins capture body motions from different segments of the body concurrently [33,34]. Therefore, motions from different body parts do not overlap, as is the case when the radar is perpendicular to the body. This separation facilitates the parallel monitoring of breathing and heartbeat motions, as the smaller heartbeat motion (0.287 to 0.568 mm [35]) can be monitored separately from the larger breathing motion (1 to 5 mm [36]). Our previous study was limited by the low number of participants and by including only healthy individuals. Because stroke can disrupt the autonomic nervous system (ANS), impairing cardiovascular functions [37,38], the performance of breathing and heart rate monitoring based on data from healthy individuals might not generalize to stroke patients.

This work aims to evaluate the use of multiple radar range bin selection in acute stroke patients sleeping in a clinical setting. The radar range bin selection was used to detect breathing and heart rates. We tested the radar technique against the gold standard of electrocardiography (ECG) and thoracic and abdominal respiratory inductance plethysmography (RIP) belts using a standard PSG. This study addresses the two main limitations of our previous study. Here we have a larger group (49 instead of 6), and instead of measuring healthy volunteers in a quiet office environment, we are measuring acute stroke patients in a clinical ward. In addition to multiple range bin selection, we also test our previously developed single bin selection approach [33] and compare the results.

## 2. Materials and Methods

### 2.1. FMCW Radar

We used the Texas Instruments (Dallas, TX, USA) IWR6843ISK-ODS FMCW radar (see [26,28,31] for FMCW radar principles in vital sign monitoring). This radar has a carrier frequency of 60 GHz, with monotonically increasing frequency modulation in a sawtooth pattern. See Table 1 for details on the radar configuration. The radar board is connected to a DCA100EVM real-time measurement board from Texas Instruments. These components are enclosed in a custom designed box (see the black box with a green sticker in Figure 1) and are connected via wires to the measurement tablet. On the tablet, we used the Lab Streaming Layer (LSL, version v1.16.2) [39] to record the data.

For computing the breathing and heart rate, we used the displacement time series. The displacement is computed in each radar range bin from the phase values. We summarized the information of each frame by taking the median phase value of the 20 chirps in one frame for each range bin. In [31], we showed the advantage of applying this chirp median method. By taking one phase value for each frame, we obtained a time series with a sampling rate of 20 Hz for each range bin.

To detect motion from the radar recording, we calculated a motion parameter based on radar Doppler FFT [40]. This parameter summarizes velocity information from an epoch composed of multiple frames. The steps for computing the motion parameter in an epoch with multiple frames are as follows:Compute the Doppler FFT for each frame in the epoch. To compute the Doppler FFT, an FFT operation is performed for each range bin along all chirps in one frame. This returns velocity information for each radar range bin.Each Doppler FFT frame is reduced from a range–velocity representation to a velocity-only representation by summing along the range bins for each velocity bin.Each velocity bin is cleaned of clutter by subtracting its median over all frames.For each frame, a Hamming window is applied to the velocity bins to reduce the influence of low-velocity bins.Finally, the velocity bins and frames are summed up to a single value.

This motion parameter highlights high velocities in the recording; therefore, it is appropriate to identify faster body movements during the recording.

### 2.2. Polysomnography Setup

We conducted sleep studies with parallel radar and PSG recordings. For PSG recordings, we used the SOMNOmedics HD eco device (Randersacker, Bavaria, Germany) with the following sensors: RIP abdominal and thorax belts, three-channel ECG, finger photoplethysmography (PPG) for pulse oximetry, three-channel electroencephalogram-electrooculogram combination (EEG-EOG) and a microphone attached to the neck. The listed sensors are connected to a small head unit, which is attached to the patient’s abdomen with the abdominal RIP belt. Our PSG setup did not include video recording.

### 2.3. Patients

The study was conducted in the post-stroke unit of the Evangelisches Krankenhaus Oldenburg (Oldenburg, Germany). The study was registered in the German Clinical Trials Register (DRKS, identifier: DRKS00028189). Patients were recruited from acute stroke patients who came through the ward after being transferred from the stroke unit. We recruited 49 patients (14 women, 35 men) who suffered an acute ischemic stroke (AIS). The median age was 68 years (minimum 45 years, minimum 86 years); the median body mass index (BMI) was 28.1 (minimum 21.3, maximum 37). The study was reviewed and approved by the medical ethics committee of the Carl von Ossietzky Universität Oldenburg (votum number 2022-016; date of approval: 14 July 2023), and all patients gave their written informed consent. Sleep-associated breathing abnormalities were not included in the inclusion criteria. The exclusion criteria included dementia, aphasic syndromes, and delirium. The ability to give consent was determined during the informational discussion by a physician.

### 2.4. Measurement Setup

We have assembled a custom radar measurement setup. See Figure 1 for a depiction of the setup at the stroke unit. The main components are the radar box and the recording tablet. The tablet is mounted on a backplate that holds two hooks to attach the setup to the bed frame. The beds were standard clinical beds, measuring 2.1 m in length. A pole mounted on the backplate holds the radar box. The radar is directed at the middle of the bed surface, while also ensuring that the entire bed is within its field of view. For handling the measurements, an intuitive graphical user interface (GUI) was created using the PyQt6 framework (version v6.5.1). The setup is powered through a power cable connected behind the patient’s bed. For the connection to the electrical grid of the clinic, we used a connector with an inbuilt circuit breaker; thus, any malfunction of the measurement setup cannot affect the clinical systems. Furthermore, the setup can easily be removed and set aside in one motion in case the patient must be transported elsewhere in an emergency.

### 2.5. Measurement Synchronization

To ensure synchronicity of the PSG and radar measurements, we have developed a custom synchronization procedure. The manufacturer of the PSG system offers an optocoupler for transmitting external signals to the PSG head unit. We used this optocoupler for synchronization. With the measurement software, we send a synchronization signal consisting of 20 markers through the optocoupler to the PSG head unit and record this signal simultaneously on the measurement tablet. The synchronization signal is a sequence of 20 square-wave signals, with each signal level change serving as a marker for synchronization. Thus, we have precisely defined time points in both recordings, which can be used in postprocessing to align all measurements to one time axis. The delay between the signal on the tablet and the PSG caused by the optocoupler was determined by a calibration measurement. Figure 2 shows the recording tablet, the PSG head unit, and the optocoupler (the small box on top of the tablet) that connects them. The same synchronization process is performed at the beginning and end of the measurement. The recordings are aligned using the markers sent at the beginning, while the markers at the end serve as an indication of whether a significant temporal drift occurred between the tablet and the PSG clock during the night.

### 2.6. Study Protocol

On the day of recording, the study personnel go to the patient between 6 and 7 p.m. First, the radar measurement setup is mounted on the bed. The tablet is started up, and the measurement personnel log into the study account. The measurement GUI starts automatically after a successful login. The GUI has only one button, which starts and stops the recording. Next, synchronization (see Section 2.5) between the radar and the PSG system is performed. The synchronization is also integrated into the GUI, guiding the personnel with a pop-up window (see Figure 2). After successful synchronization, the user account on the radar measurement tablet is locked for the night. The optocoupler is disconnected from the PSG system for the duration of the recording. The personnel then attach the PSG sensors to the patient. From then on, the patient can freely move around in the clinic and decide when to go to sleep. When sleeping, the patient is free to choose their sleeping posture. The next morning, the study personnel go to the patient at 7 a.m. First, they remove the PSG sensors. The measurement is then stopped, and the end synchronization is completed. The radar recording is automatically saved to the tablet.

### 2.7. Radar Range Bin Selection

For single bin selection, the breathing and heartbeat range bins are ranked using temporal phase coherence (TPC) [32]. From the range bins with the five highest TPC values, breathing and heart rates are computed. The final rate for the given epoch is computed from the range bin that has the highest quality breathing or heartbeat signals from the preselected group of five (see details in [33]).

The rationale for multiple over single range bin selection for computing breathing and heart rates is that the summarization of information from multiple spatial sources makes the computation more robust. The method for multiple bin selection is also considerably stricter in its choices, reducing the chance of false positives. Because single bin selection only performs a ranking using TPC, there is a higher chance that incorrect bins get through. This difference in range bin selection was explored in detail in [31].

The methodology for multiple bin selection is based on persistence homology [41,42], exploiting the fact that breathing and heartbeat signals have a specific shape in the radar phase signal. We chose this complex method for range bin selection because of the low specificity of more classical approaches, such as spectrum- or autocorrelation-based range bin selection. Although these classical approaches can often identify valid breathing and heartbeat signals, they are not reliable in rejecting unrelated signals that happen to share spectral characteristics. When developing persistence homology-based bin selection, we empirically derived thresholds for evaluating persistence diagrams computed from radar signals. In this article, we evaluate how well these settings generalize to a larger dataset from a different population in a different setting. For a detailed description of the algorithms, we refer the interested reader to the original publication [31].

### 2.8. Breathing and Heart Rate Computation

Breathing rates were computed using an autocorrelation-based method, while heart rates were computed based on peak detection. The reference rates were computed from the PSG sensors using the same approaches. We have described the details of these algorithms in [33]. The algorithms were implemented in the Python programming language (version 3.12.2) [43]. The computations were executed offline using a computing server with two Intel (Santa Clara, CA, USA) Xeon Gold 6240 CPUs and 768 GB of RAM.

### 2.9. Adaptations in the Algorithms

Compared to the original implementation [31], we adapted two areas. First, we adjusted the expected breathing rate and heart rate intervals, meaning the minimal and maximal rates that the algorithm considers. This change was based on our observation that the reference sensors showed breathing and heart rates exceeding the boundaries we used previously. We changed the breathing rate boundaries from 10–22 breaths per minute (BPM) to 6–30 BPM, which covers the expected breathing rates in a more elderly population [44]. The heart rate boundaries were originally 40–120 beats per minute (BPM); here, we used 40–150 BPM. This new interval covers normal heart rate ranges [45] and also accounts for sudden increases in heart rate.

Second, we targeted the slow running time of the multiple bin selection algorithm that we have described in [31]. We implemented CPU parallelization at multiple stages of the algorithm using the *joblib* Python library (version 1.4.0). For timing the algorithms, we used the *perf_counter* function of the Python *time* module.

### 2.10. Description of the Dataset

The 49 sleep recordings comprise a total duration of 611 h and 26 s. The goal of this study was to compute breathing and heart rates during sleep. Due to the study protocol (see Section 2.6), the recordings include long intervals during which the patients were not asleep. To focus the analysis on intervals when patients were most likely to be asleep, we restricted the recordings to the time window between 10 p.m and 6 a.m. This reduced the total duration of the recordings to 394 h, 2 min, and 35 s.

Furthermore, we excluded intervals with high levels of motion using the motion parameter described in Section 2.1. Periods in which the level of movement exceeds the overnight average are indicative of wakefulness [46]. To identify such intervals, we computed the motion parameter in 60 s epochs with a 55 s overlap. Next, we applied a moving-average filter with a window length of 5 min and then calculated the average motion level over the entire recording, removing all segments exceeding this threshold. After excluding high-motion intervals, the total duration of the 49 recordings was further reduced to 244 h, 15 min, and 25 s.

Breathing and heart rates were computed after segmenting the recordings into epochs. Each patient’s recording was split into 60 s epochs, with a 55 s overlap, providing fine temporal resolution of the breathing and heart rate variations throughout the night. Both single bin selection and multiple bin selection methods were applied using this epoching scheme. This represents a change from our previous publication [31], where single bin selection used a 40 s overlap, consistent with the original publication [33]. Using the same 55 s overlap for both methods facilitates a more interpretable comparison.

### 2.11. Evaluation Metrics

To evaluate the performance of the breathing and heart rate computations, we used the following metrics: mean absolute error (MAE)(1)$$\mathrm{MAE}=\frac{1}{N}\sum_{n=1}^{N}\left|f_{PSG}\left(n\right)-f_{Radar}\left(n\right)\right|,$$
and mean absolute percent error (MAPE)(2)$$\mathrm{MAPE}=100\cdot \frac{1}{N}\sum_{n=1}^{N}\frac{|f_{PSG}\left(n\right)-f_{Radar}\left(n\right)|}{f_{PSG}\left(n\right)},$$
where *N* is the total number of epochs, while $f_{Radar}\left(n\right)$ and $f_{PSG}\left(n\right)$ correspond to the breathing or heart rates of the radar and PSG of the *n*th epoch, respectively.

In addition to the error metrics, we also computed Spearman’s correlation coefficient to express how closely the radar breathing and heart rates follow the reference rates. Spearman’s rank correlation is defined as(3)$$\rho_{s}=\frac{cov(R(x),R(y))}{\sigma (R(x))\cdot \sigma (R(y))},$$
where R(x) is the rank of the variable, cov(R(x),R(y)) is the covariance of the rank, and σ(R(x)) is the standard deviation of the rank.

We also report the ratio and count of epochs where the algorithms successfully computed the breathing and heart rates (hereafter referred to as recall). In addition, we calculated the duration for which no rate was computed. This is important to add to the previously mentioned metric because, given the overlap between epochs, it is not trivial to determine what the actual coverage of computed rates on the recordings is. The final metric is the ratio of epochs in which the radar breathing or heart rate error (difference to the PSG sensor) is lower than ±1 BPM. This ratio is reported in relation to the epochs in which both reference and radar rates were successfully computed.

## 3. Results

We computed the breathing and heart rate results from the recordings of 49 patients, totaling 244 h, 15 min, and 25 s. In total, there were 175,425 epochs. We measured the running time of the algorithms with CPU parallelization, including epoching and computation. We excluded the time for loading the recording, as that varied depending on the server load. The average running time per patient for the breathing rate multiple bin selection algorithms was 1 h, 3 min, and 7 s. The heartbeat multiple bin selection took an average of 25 min and 43.9 s to compute. With single bin selection, the average running time per patient for the breathing rate calculation was 12 min and 59.2 s, while for the heart rate calculation, it was 28 min and 13 s.

### 3.1. Distribution of Selected Range Bins

Figure 3 shows the distribution of the selected range bins for breathing (top) and heartbeat (bottom) using single (left) and multiple (right) bin selection. These plots highlight the different spatial segments in which the algorithms identify the breathing and heartbeat signals.

The selected range bins vary between patients, but there is a distinct distribution for breathing and heartbeat. The breathing distribution starts farther away from the radar compared to that of the heartbeat. Furthermore, the peak of the breathing range bins’ distribution is wider.

The single and multiple bin selection distributions have a similar shape, both for breathing and heartbeat. There are only slight differences, such as the breathing multiple selection distribution having a wider peak and the heartbeat multiple selection distribution having a more prominent peak than the single selection. The overall counts are higher for the multiple selection distributions, as it can select more than one range bin for each epoch.

### 3.2. Breathing Rate

Table 2 shows the comparison of the breathing rate computation results for single and multiple bin selection. With single bin selection, the recall for the radar breathing rate was 93.49%, while with multiple bin selection, it was 73.38%. In terms of errors, multiple bin selection outperformed single bin selection, with MAE and MAPE values 2.2 and 2.3 times lower, respectively. Furthermore, in the epochs in which rates were calculated for both reference and radar, 8.36% more of those epochs had lower than ± 1 BPM errors. Multiple bin selection had a higher correlation with the reference values than single bin selection (0.95 vs. 0.85). Figure 4 presents the per-patient results using box plots, with each patient’s result individually represented by gray circles. The values of the box plots are presented in tabular format in the Appendix A.

Figure 5 shows Bland–Altman plots of the breathing rate results from single (left panel) and multiple (right panel) bin selection. The mean breathing rate error with single bin selection is −0.20 BPM, while with multiple bin selection it is 0.04 BPM. The 95% confidence interval of the errors is approximately twice as wide with single bin selection ([−4.87, 4.47] vs. [−2.31, 2.39]). For single bin selection, 3.31% of the points are below the lower limit of the confidence interval, and 1.67% are above the upper limit. For multiple bin selection, these values are 1.49% and 1.54%. The distributions of the plots show no appreciable differences. Both are symmetric and do not show any obvious trends.

The overnight breathing rates for the same example patient are shown in Figure 6 and Figure 7, calculated using single and multiple bin selection, respectively. In both figures, the upper panel shows the breathing rates computed from the reference and radar. The middle panel shows the error of the radar breathing rates. The lower panel shows the range profile; that is, the range bins that were selected for the computation of the breathing rate. The gray shading in all panels shows the segments that were disregarded because of high levels of motion.

### 3.3. Heart Rate

Table 3 shows the result metrics for the heart rate calculation. The heart rate results show the same pattern as the breathing rates, although with larger differences. The recall is 4.1 times higher with single bin selection. However, the MAE and MAPE values are 4.8 and 5.3 times lower with multiple bin selection, respectively. Among the epochs in which both reference and radar heart rates were calculated, the ratio of errors lower than ±1 BPM is 1.6 times higher with multiple selection. Finally, heart rates computed with multiple bin selection have a 1.7 times higher correlation coefficient with the reference rates. Figure 8 presents the per-patient results with box plots, with gray circles representing individual patients. The values of the box plots are presented in tabular format in the Appendix A.

The Bland–Altman plots for the heart rates computed with single and multiple bin selection are shown in Figure 9. Multiple bin selection (right panel) has a mean of −0.43 BPM, while single bin selection (left panel) has a mean of −3.05 BPM. The single bin selection 95% confidence interval is more than three times as wide ([−17.83, 11.73] vs. [−4.64, 3.78]). For single bin selection, 7.26% of the points are below the lower limit of the confidence interval, and 0.24% are above the upper limit. For multiple bin selection, these values are 1.68% and 0.40%. Both approaches have an overall negative bias and show a similar trend: with an increasing heart rate, the radar goes from underestimating to overestimating the reference rate. However, the distribution of the plots is noticeably different, especially with single bin selection having more values underestimating the reference rate.

Figure 10 and Figure 11 show an example night of computed heart rates for the same patient, using single and multiple bin selection, respectively. The results shown are from the same patient as the breathing rate results in Figure 6 and Figure 7. The layout of the panels is the same as in those figures. Gray shading shows periods with high levels of motion based on the radar motion parameter. The negative bias pointed out in Figure 9 is visible with more negative difference values present in the middle panel of the two overnight figures.

## 4. Discussion

In this article, we have evaluated two approaches for radar-based breathing and heart rate calculation in a clinical dataset of acute stroke patients. We have used a single- and multiple range bin selection approach from our previous work, with only minor modifications. Previously, we evaluated these methods only in a small sample of six healthy volunteers who slept in an office environment. This current study included 49 acute stroke patients sleeping in a clinical environment, which provides a better perspective on the robustness of the methods.

### 4.1. Performance of Algorithms

Multiple bin selection led to lower error values for both breathing and heart rate. The MAE and MAPE for the breathing were 0.39 BPM and 2.48%, while for the heart rate they were 0.84 BPM and 1.44%, respectively. These values are slightly worse than those in the healthy volunteer dataset (breathing rate MAE 0.29 BPM, MAPE 2.24%; heart rate MAE 0.66 BPM, MAPE 1.02%).

The results in the clinical dataset show similar patterns as in healthy volunteers, but with more exaggerated differences between the single- and multiple bin selection approaches. Single bin selection has a higher recall for both the breathing and heart rates, especially the heart rates, where there is a 4.1 times difference. For the breathing rate, this difference is comparatively small, with a factor of 1.3 difference. On the other hand, both the computed breathing and heart rates have substantially lower errors using multiple bin selection. For the breathing rates, the MAE is 2.2 times lower, while for the heart rates, it is 4.8 times lower.

There is a large variation in the results between individual patients, as shown by the box plots in Figure 4 and Figure 8. In particular, the recall values with multiple selection span a wide range, even disregarding extreme outliers. For breathing rates computed with multiple selection, the difference between the 10th and 90th percentiles was 53.35 percentage points (see Appendix A). For heart rates computed with multiple selection, the same difference was 38.55 percentage points (see Appendix A). Single bin selection performed more consistently in recall; however, the error and correlation values did show large variations. The Spearman correlation coefficient for heart rates with single bin selection had 10th and 90th percentiles of −0.09 and 0.93, respectively (see Appendix A).

### 4.2. Comparison with the Literature

To place our results in context, we searched the literature for related publications. Stroke disrupts the ANS [37,38], which can affect cardiovascular function, making monitoring more challenging. Therefore, our initial aim was to compare our findings with studies involving stroke patients. However, we did not find any studies on radar-based sleep monitoring specifically in stroke populations. Therefore, we expanded our search to recent studies that used radar-based overnight monitoring of breathing or heart rate, regardless of the study population. We identified five relevant works [47,48,49,50,51]. Table 4 summarizes these studies, including the radar type, study population, protocol, and reported breathing and heart rate results. For consistency, we extracted only the metrics that were also reported in our study.

Compared with studies reporting breathing rate results [47,48,50,51], our multiple bin selection approach achieved a lower MAE and MAPE, as well as a higher correlation. However, the limits of agreement (the limits of the 95% confidence interval) in our Bland–Altman analysis were wider than those reported in three of these studies [48,50,51].

Only two of the five studies reported heart rate results [49,51], and both provided only a subset of the metrics we reported. In [49], the authors validated their method using data from 62 healthy individuals. Their reported heart rate accuracy was lower than that achieved with our multiple bin selection but higher than that achieved with our single bin selection. However, the recall was higher than in both of our approaches. The study in [51] included elderly participants, both healthy individuals and patients with neurodegenerative diseases. However, the authors reported only correlation and Bland–Altman limits of agreement. Based on these metrics, our multiple bin selection performed better, whereas our single bin selection performed worse.

### 4.3. Discussion of Key Findings

The contrast between the recall and error values highlights the fundamentally different characteristics of the two range bin selection methods. Single bin selection is more permissive in computing breathing and heart rates and, therefore, yields a higher proportion of epochs with estimated rates, albeit at the cost of larger errors. Multiple bin selection, in contrast, applies stricter criteria when selecting range bins. As a result, the recall is lower, particularly for the estimation of the heart rate (see [31] for an analysis of range bins rejected by the multiple bin selection method). However, when a rate is calculated, it is highly accurate, as reflected by the low MAE and MAPE values of the radar heart rates. For 75% of patients, the heart rate MAE is below 1.34 BPM and the MAPE below 2.28% (see Figure 8, panels C and D and Appendix A).

The heart rate computation with single bin selection has a higher recall and 2.5 times more epochs with errors lower than ±1 BPM. However, as panels B, C, and D in Figure 8 and the Bland–Altman plot in Figure 9 demonstrate, there are too many recordings in which the estimate is too unreliable. This demonstrates the inability of range bin selection solely based on ranking to discard signals where no accurate heart rate can be computed. A possible improvement would be to integrate the ranking produced by the TPC [32,33] into the decision step based on persistence homology [31]. This would combine the high recall of TPC with the specificity of persistence homology.

There are two factors independent of the algorithms that contribute to low heart rate recall with multiple bin selection (19.93%). Heartbeats are, in the best case, half the size of breathing motions but can be approximately 17 times smaller (0.287–0.568 mm [35] vs. 1–5 mm [36]). This makes accurate detection difficult even in an optimal setting, such as our study with healthy volunteers in a quiet office room [31,33]. In addition, the ANS of stroke patients is disrupted [37,38]. This can impact cardiovascular functioning in an inhomogeneous way, depending on the exact injury of the patient. The low recall rates, together with these two factors, point to the need for specifically designed heart rate computation algorithms for stroke patients. A general approach that can be fine tuned for individuals would address the inhomogeneity in stroke injuries.

In calculating breathing rates, multiple bin selection had a more comparable recall to single bin selection (73.38% vs. 93.49%); however, as panel A in Figure 4 shows, there was a large variation among patients. Given the more reliable accuracy of multiple selection (see panels B, C, and D of Figure 4) and the additional information provided by the range profile (see Figure 7), it presents a better option for monitoring breathing rates than single bin selection.

The distribution of selected range bins in this clinical dataset shows patterns similar to those observed in the healthy volunteer dataset. The breathing range bins are located farther from the radar than the heartbeat range bins and also span a wider spatial region. This indicates that the tendency of multiple range bin selection to associate breathing with range bins closer to the upper body and heartbeat with bins closer to the lower body is not dataset dependent. The clear spatial separation between the breathing and heartbeat range bins supports the use of lateral radar placement, specifically at the foot end of the bed. By achieving intrinsic separation of the signals, this configuration avoids the need for complex signal separation algorithms that are otherwise required when breathing and heartbeat signals overlap within the same range bin.

The implementation of CPU parallelization reduced the runtime by approximately a factor of five compared to the original implementation [31]. The observed runtime differences between single and multiple range bin selection reflect the differing computational complexity of the underlying algorithms. For multiple bin selection, we used two persistence homology methods: Vietoris–Rips filtration and sublevel set filtration [41]. Vietoris–Rips filtration is computationally intensive, while sublevel set filtration has a complexity comparable to the TPC method [32] used for single bin selection [33]. During breathing rate estimation, multiple range bin selection applies both filtration methods, while for heart rate estimation, only sublevel set filtration is used. This explains why multiple bin selection for breathing rate estimation is approximately five times slower than single bin selection, whereas for heart rate estimation, the runtimes are comparable.

### 4.4. Potential of the Method

Accurate breathing and heart rates, especially when using multiple bin selection, show promise in the use of radar-based sleep monitoring for at-risk patients in clinical wards. The contactless nature of the radar avoids the burden of contact-based sensors on patients and caregivers while still providing useful insights into the patients’ states. Although the pipelines introduced here only provide breathing and heart rates and not further evaluations, such as the detection of abnormal respiratory patterns or the classification of sleep stages, this information is already valuable for assessing patients’ sleep. Breathing and heart rates outside the expected boundaries can indicate underlying disease. In addition, the overnight evolution of the rates can also provide caregivers with insights into whether there were many sleep interruptions or if the patient could sleep peacefully.

Having a good range bin selection method should also enable the detection of abnormal respiratory patterns, such as sleep apnea and hypopnea [52]. Sleep apnea or hypopnea occurs when a patient’s airflow stops or is reduced. Therefore, if we have a good method for detecting breathing, that should be a good starting point to look for cessations in breathing. In particular, central events, where airflow stops or is reduced due to a lack of breathing effort, should be detectable with this approach.

### 4.5. Limitations

The sample of 49 patients provides a better basis for evaluating the efficacy of our methods than the six healthy volunteers whose data we tested before [31,33]. However, given the specific condition of the patients in this study—acute ischemic stroke patients—it is possible that the results do not reflect the method’s performance in the general population, although we assume that the general population would have calmer sleep than this patient population, which should lead to better performance, not worse. Furthermore, the sexes in this study were not balanced (14 female, 35 male); therefore, the results might be less representative for monitoring the sleep of women.

We have cut each recording to focus on a typical sleeping window. In addition, we used a radar derived motion parameter to remove periods with high levels of motion. Although these are sensible steps to clean the dataset before computing breathing and heart rates, a more sophisticated set of algorithms might do a better job of keeping only sleep epochs for each individual. A bed presence detector with a more complex motion parameter should be implemented together with a dedicated sleep staging algorithm for optimal sleep monitoring.

The manual setting of parameters for the range bin selection algorithms is another limitation. A machine learning based approach to classifying range bins based on persistence homology features might provide a more optimal solution. The large hurdle for such an approach is the lack of clear labels for the range bins that the machine learning model could learn.

By setting the lower boundary of the expected breathing rate range to 6 BPM (see Section 2.9), there is a chance of treating apneas as normal breathing cycles. While the algorithms presented in this article are not meant for detecting apneas, it is still important to address this possibility. The breathing rate computation algorithm itself has a check that controls how consistent the breathing cycles are in an epoch (see [33] for details). Therefore, in epochs with apnea events, no breathing rate should be returned. However, if an apnea detection algorithm were to be combined with this breathing rate algorithm, it is an important case to investigate by comparing epochs where breathing is detected against expert apnea annotations.

The last limitation we would like to point out is that the method was not tested on more accessible computing platforms. To efficiently process the 2.5 TB of raw data in our study, we used a powerful computing server, and with CPU parallelization, we were able to achieve manageable running times. The algorithms we used are compatible with widely available modern computers for the intended offline evaluation. Although large file sizes might require users to process recordings in shorter segments instead of the whole night at once.

### 4.6. Outlook

We will investigate whether a combination of the single- and multiple range bin selection algorithms can improve the heart rate calculation recall in this dataset. Then, our next focus will be on developing a set of algorithms to investigate the potential of radar-based contactless screening of sleep apnea and hypopnea in acute stroke patients in a clinical setting. We will evaluate whether the range bin selection tested here can be successfully integrated into the detection of respiratory events. A simple-to-apply method in neurology clinics would be very beneficial for screening sleep apnea and hypopnea to aid in the better recovery of stroke patients.

## 5. Conclusions

We have demonstrated the performance of multiple range bin selection for radar-based breathing and heart rate computation in sleeping acute stroke patients. In terms of accuracy, multiple selection outperforms the more commonly used single selection. However, single selection has higher temporal coverage, especially for heart rates. Therefore, single bin selection could be a useful supplement, especially for heart rate computation. These results are a valuable contribution to developments that incorporate radar-based sleep monitoring in clinical wards in the future.

## Figures and Tables

**Figure 1 sensors-26-00251-f001:**
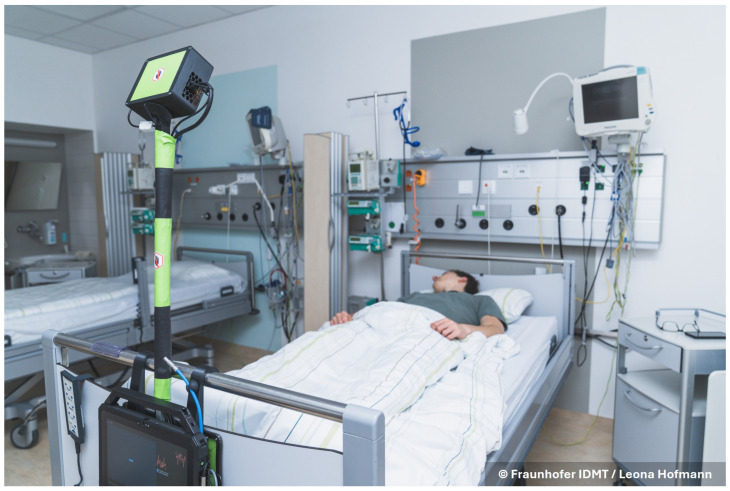
The measurement setup mounted on a clinical bed in the stroke unit. The setup itself is in the foreground. The measurement tablet is on the bottom, showing the graphical user interface. Above the tablet, the hooks holding the setup are visible. The radar box is held up by a pole which also leads the cables to the radar.

**Figure 2 sensors-26-00251-f002:**
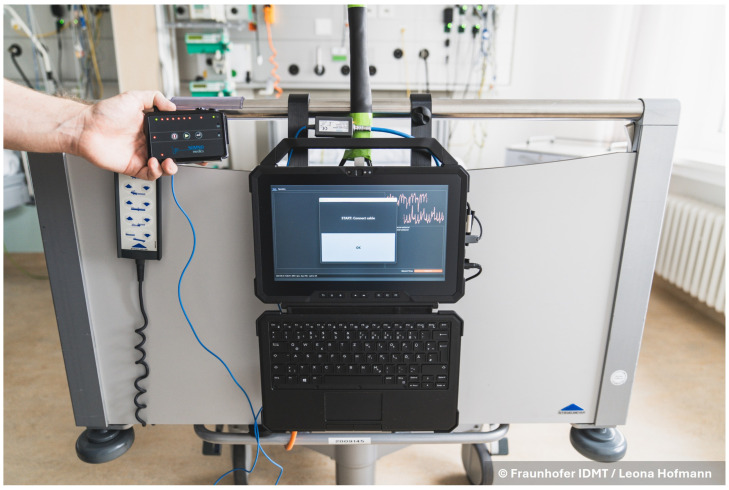
The graphical user interface of the measurement setup. The photo shows the process of synchronization, with the PSG head unit (held in the hand) connected to the measurement tablet (in the middle) over the manufacturer provided optocoupler (above the tablet).

**Figure 3 sensors-26-00251-f003:**
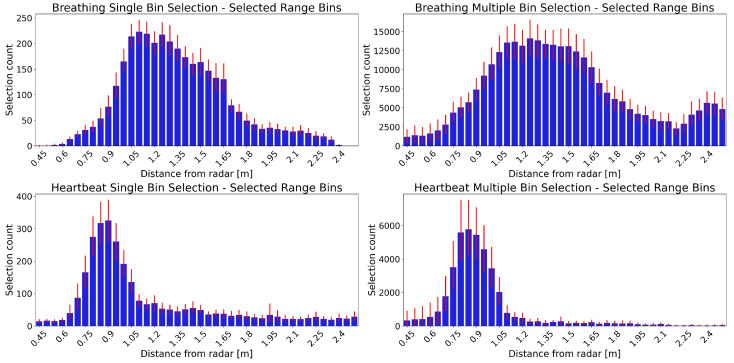
Distribution of selected range bins from the results of 49 patients. The horizontal axis corresponds to the range bins, starting on the left with those closer to the radar. The vertical axis shows how many times a range bin was selected. The blue bars show the mean selection count for each range bin over all 49 patients. The red error bars show the 95% confidence interval. Panels in the **upper** row show breathing selections, while those in the **lower** row show the heartbeat range bin selections. The distribution of range bins selected with single bin selection are in the **left** panels, while those selected with multiple bin selection are in the **right** panels.

**Figure 4 sensors-26-00251-f004:**
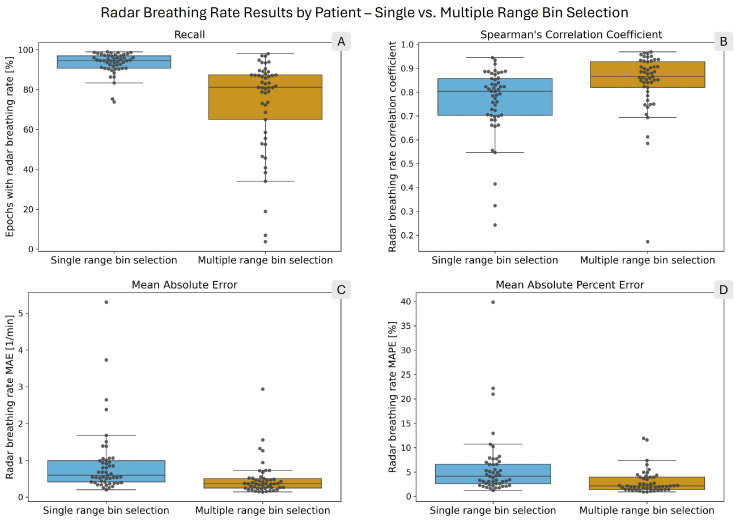
Box plots showing the distribution of per-patient radar breathing rate performance metrics. The overlaid points each correspond to one patient. In each panel, the blue box on the left shows the single range bin selection results, while the orange box on the right shows the multiple range bin selection results. (**A**) shows the radar breathing rate recall. (**B**) presents Spearman’s correlation coefficient. (**C**) shows the mean absolute error (MAE), and (**D**) shows the mean absolute percent error (MAPE) of the computed radar breathing rates.

**Figure 5 sensors-26-00251-f005:**
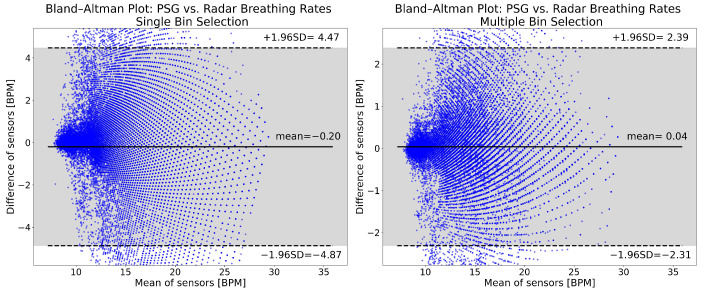
Breathing rate Bland–Altman plots for single (**left panel**) and multiple bin selection (**right panel**). The plots were made from the data of all 49 patients, after cutting the recordings and removing high motion intervals. The horizontal axis shows the mean of the reference and radar breathing rates. On the vertical axis, the difference between the computed breathing rates is shown. Both are expressed in breaths per minute (BPM). The solid horizontal line in the middle is the mean of differences, while the two dashed lines show the upper and lower limits of the 95% confidence interval. All of the values indicated are in BPM. The 95% confidence interval is also indicated with the gray shading.

**Figure 6 sensors-26-00251-f006:**
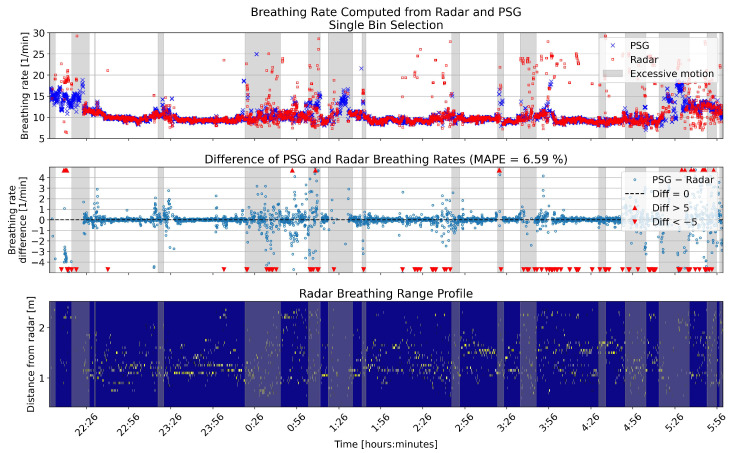
One example full night comparison of breathing rates computed from the PSG sensor and radar single bin selection. The **upper** panel shows the breathing rates in breaths per minute (BPM) for the two sensors. In the **middle** panel, the differences between the rates are shown in BPM. The **lower** panel shows the range profile, that is, the range bins that were selected for breathing rate computation (marked with yellow). Gray shading in each panel shows the segments that had high levels of motion based on the radar motion parameter.

**Figure 7 sensors-26-00251-f007:**
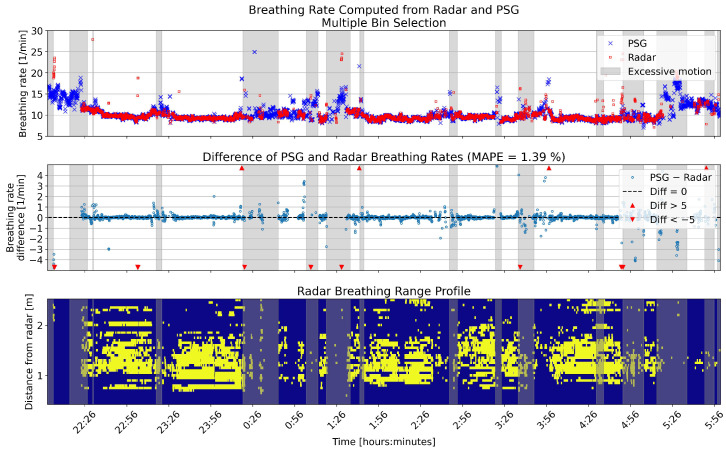
One example full night comparison of breathing rates computed from the PSG sensor and radar multiple bin selection. The **upper** panel shows the breathing rates in breaths per minute (BPM) for the two sensors. In the **middle** panel, the differences between the rates are shown in BPM. The **lower** panel shows the range profile, that is, the range bins that were selected for breathing rate computation (marked with yellow). Gray shading in each panel shows the segments that had high levels of motion based on the radar motion parameter.

**Figure 8 sensors-26-00251-f008:**
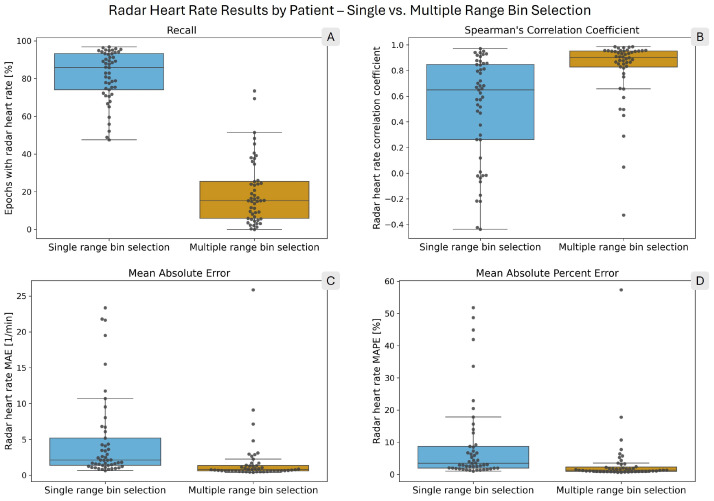
Box plots showing the distribution of per-patient radar heart rate performance metrics. The overlaid points each correspond to one patient. In each panel, the blue box on the left shows the single range bin selection results, while the orange box on the right shows the multiple range bin selection results. (**A**) shows the radar heart rate recall. (**B**) presents Spearman’s correlation coefficient. (**C**) shows the mean absolute error (MAE), and (**D**) shows the mean absolute percent error (MAPE) of the computed radar heart rates.

**Figure 9 sensors-26-00251-f009:**
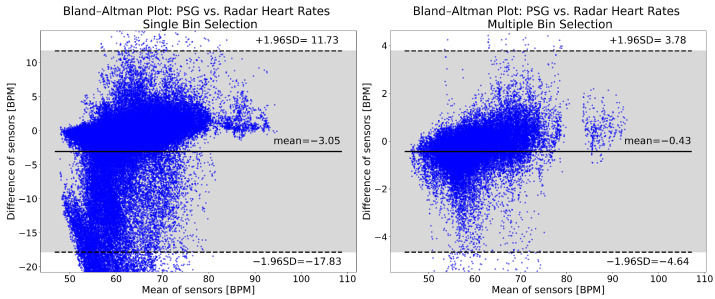
Bland–Altman plots of heart rates computed from the reference PSG and the radar. The **left** panel shows the single bin selection results, while the **right** panel shows the multiple bin selection results. The plots show data from 49 patients, after cutting the recordings and removing high motion intervals. The mean of the reference and radar heart rates is shown on the horizontal axis. The difference between the reference and the radar heart rates is represented by the vertical axis. Both are expressed in beats per minute (BPM). The solid horizontal line in the middle is the mean of differences, while the two dashed lines show the upper and lower limits of the 95% confidence interval. All of the values indicated are in BPM. The 95% confidence interval is indicated also with the gray shading.

**Figure 10 sensors-26-00251-f010:**
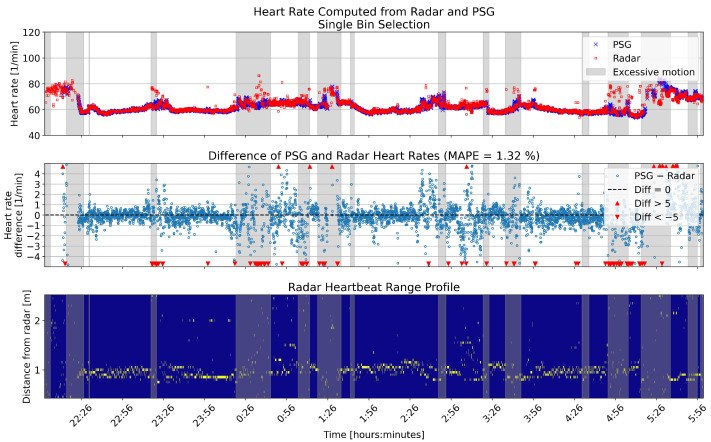
Overnight heart rates computed with single bin selection from one example patient, compared to the reference. The **upper** panel shows the heart rates in beats per minute (BPM) for the two sensors. In the **middle** panel, the differences between the rates are shown in BPM. The **lower** panel shows the range profile, that is, the range bins that were selected for heart rate computation (marked with yellow). Gray shading in each panel shows the segments that had high levels of motion based on the radar motion parameter.

**Figure 11 sensors-26-00251-f011:**
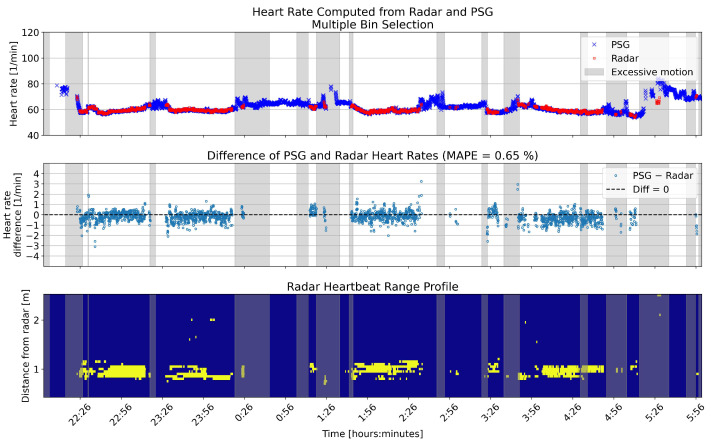
Overnight heart rates computed with multiple bin selection from one example patient, compared to the reference. The **upper** panel shows the heart rates in beats per minute (BPM) for the two sensors. In the **middle** panel, the differences between the rates are shown in BPM. The **lower** panel shows the range profile, that is, the range bins that were selected for heart rate computation (marked with yellow). Gray shading in each panel shows the segments that had high levels of motion based on the radar motion parameter.

**Table 1 sensors-26-00251-t001:** Radar configuration.

Frequency band	60–63 GHz
Range resolution	0.05 m
Azimuth and elevation FoV	120°
ADC samples per chirp	64
Chirp count per frame	20
Frame rate	20 Hz
Number of transmit antennas used	1
Number of receive antennas used	1
Maximal range of detection	2.56 m

FoV: field of view; ADC: analog to digital converter.

**Table 2 sensors-26-00251-t002:** Breathing rate results, computed from 49 overnight recordings. Comparison between single and multiple radar range bin selection.

	Single Range Bin Selection [33]	Multiple Range Bin Selection [31]
PSG recall [% (count)]	93.71 (164,393)	
Radar recall [% (count)]	93.49 (163,999)	73.38 (128,728)
Duration where no PSG rate was computed [hh:mm:ss]	00:28:25	
Duration where no radar rate was computed [hh:mm:ss]	02:19:30	35:35:35
Epochs with <±1 BPM error (for epochs with PSG and radar rate computed) [% (count)]	84.90 (132,680)	93.26 (116,508)
Mean absolute error [1/min]	0.87	0.39
Mean absolute percent error [%]	5.71	2.48
Spearman’s correlation coefficient	0.85	0.95

BPM: breaths per minute.

**Table 3 sensors-26-00251-t003:** Heart rate results, computed from 49 overnight recordings. Comparison between single and multiple radar range bin selection.

	Single Range Bin Selection [33]	Multiple Range Bin Selection [31]
PSG recall [% (count)]	90.77 (159,238)	
Radar recall [% (count)]	81.85 (143,582)	19.93 (34,963)
Duration where no PSG rate was computed [hh:mm:ss]	13:56:05	
Duration where no radar rate was computed [hh:mm:ss]	02:51:50	154:39:40
Epochs with <±1 BPM error (relative to epochs with PSG and radar rate computed) [% (count)]	50.94 (67,291)	80.09 (26,641)
Mean absolute error [1/min]	3.99	0.84
Mean absolute percent error [%]	7.67	1.44
Spearman’s correlation coefficient	0.56	0.96

BPM: beats per minute.

**Table 4 sensors-26-00251-t004:** Comparison of studies presenting radar-based breathing and heart rate monitoring during sleep. For the sake of comparison only the result metrics that were also included in our work are presented. The bottom two rows show our results with single and multiple range bin selection.

Study	Radar Type	Subjects	Study Protocol	Breathing Rate Results	Heart Rate Results
Do et al. 2022 [47]	FMCW 60 GHz (Albus Home^TM^)	*N* = 32, 8 of them children; both healthy individuals and patients with chronic respiratory condition	Overnight measurements at home, 10 subjects had 2 h evaluated, others only 15 min	MAE = 0.6 BPM	-
Bujan et al. 2023 [48]	Doppler 24 GHz (Sleepiz One+)	*N* = 139; patients with suspected sleep apnea	Overnight measurements in sleep laboratory	MAE = 0.48 BPM LoA = [−0.88, 1.61]	-
Xu et al. 2023 [49]	FMCW 60 GHz (Google Nest Hub)	*N* = 122, 62 used for evaluation; healthy individuals	Overnight recordings	-	MAE = 1.69 BPM MAPE = 2.67% Recall = 88.53%
Ravindran et al. 2024 [50]	IR-UWB (Somnofy; VitalThings)	*N* = 17; elderly individuals, those with stable comorbidities were included	Overnight recordings in sleep laboratory	MAPE = 4.64% LoA = [−2.48, 1.47]	-
Yin et al. 2025 [51]	UWB (XeThru X4; Novelda)	*N* = 47; elderly participants, both healthy individuals and patients with neurodegenerative diseases	Overnight sleep recording in a research facility	r = 0.88 LoA = [−1.2, 1.2]	r = 0.65 LoA = [−6.7, 4.9]
Our results SRBS	FMCW 60 GHz (Texas Instruments)	*N* = 49; acute ischemic stroke patients	Overnight sleep recording in post-stroke unit	MAE = 0.87 BPM MAPE = 5.71% LoA = [−4.87, 4.47] r = 0.85 Recall = 93.49%	MAE = 3.99 BPM MAPE = 7.67% LoA = [−17.83, 11.73] r = 0.56 Recall = 81.85%
Our results MRBS	FMCW 60 GHz (Texas Instruments)	*N* = 49; acute ischemic stroke patients	Overnight sleep recording in post-stroke unit	MAE = 0.39 BPM MAPE = 2.48% LoA = [−2.31, 2.39] r = 0.95 Recall = 73.38%	MAE = 0.84 BPM MAPE = 1.44% LoA = [−4.64, 3.78] r = 0.96 Recall = 19.93%

LoA: Bland–Altman plot limits of agreement, the 95% confidence interval; IR-UWB, UWB: impulse-radio ultra-wideband; r: correlation coefficient; SRBS and MRBS: Single and multiple range bin selection.

## Data Availability

The raw data supporting the conclusions of this article will be made available by the authors upon request, in accordance with the consent of the test subjects as per the data protection agreement. To avoid misunderstandings, we would like to point out that the radar system used here is a demonstrator exclusively for research and development purposes (research use only). It is not CE-certified as a medical device, and no performance evaluation has been carried out in accordance with the Medical Device Regulations. Data and analyses are not used in diagnostic or therapeutic decisions. The overnight plots of breathing and heart rate results for both single and multiple range bin selection for each patient are available upon request. The source codes used for computing the results presented in this article can be found in this repository: https://doi.org/10.5281/zenodo.17559602.

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
