# Peer review of "Radar Multiple Bin Selection for Breathing and Heart Rate Monitoring in Acute Stroke Patients in a Clinical Setting"

_sensors, 2025, doi:10.3390/s26010251_

Round 1

Reviewer 1 Report

Comments and Suggestions for Authors

The authors aim to evaluate a novel radar signal processing method, termed "multiple bin selection," for monitoring breathing and heart rate in a clinically relevant population of acute stroke patients. According to a rigorous comparison against the gold standard, Polysomnography (PSG), the proposed algorithm is demonstrated to significantly enhance measurement accuracy over a traditional "single bin selection" approach, particularly for the more challenging task of heart rate estimation. The study provides a valuable real-world validation of a promising contactless monitoring technology. However, several critical limitations and unclear definitions degraded the quality of the paper. My comments are as follow.

  1. The extremely low data coverage for heart rate estimation severely compromises the proposed method's clinical utility. The most significant limitation is that the multiple bin selection method yields a valid heart rate in only 15.05% of the recorded epochs. For clinical applications, such as the detection of nocturnal arrhythmias or the assessment of autonomic nervous system function, a monitoring system that is inoperative for 85% of the night is practically unusable. While the manuscript acknowledges this accuracy-coverage trade-off, it fails to present a validated strategy to mitigate this crucial shortcoming. This omission significantly detracts from the translational value of the reported findings.
  2. The manuscript lacks a mechanistic explanation for the disproportionate signal loss in heart rate versus breathing rate. The data reveals a stark disparity: transitioning to the stricter algorithm reduces breathing rate coverage by a factor of 1.4, whereas heart rate coverage plummets by a factor of 4.9. The authors do not offer a substantive physiological or physical explanation for this discrepancy. Without this analysis, it is impossible for the reader to discern whether the difficulty in acquiring heart rate signals stems from (a) the specific pathophysiology of stroke patients (e.g., poor peripheral perfusion, limb tremors), (b) environmental factors in the clinical ward, or (c) a fundamental inadequacy of the algorithm in detecting minute ballistocardiogram movements. This lack of mechanistic insight diminishes the scientific depth of the paper and offers no clear direction for future algorithm optimization.
  3. The methodology for defining the analysis period is subjective and lacks reproducibility. The authors state in Section 2.10 that the recordings were "manually inspected" to identify and select the sleep periods for analysis. This manual procedure introduces a significant risk of subjective bias and compromises the reproducibility of the study. Without explicit, objective criteria for this data-cutting process, or a reported measure of inter-rater reliability, the validity of the entire performance evaluation is undermined, as the very foundation of the analysis—the dataset itself—is not robustly defined.
  4. An anomalous finding regarding computational time is reported without explanation. The results section presents a counter-intuitive observation: the more computationally complex multiple bin selection algorithm ran faster for heart rate estimation (30 min) than the simpler single bin selection method (35 min). This unexplained anomaly could lead readers to question the rigor of the experimental execution or the accuracy of the results reporting. A plausible explanation—that the algorithm "terminates early" in the vast majority of epochs due to poor signal quality—is not provided by the authors. Clarifying this point is essential for maintaining the reader's confidence in the study's technical details.
  5. The abstract and conclusions present a biased perspective by omitting the critical finding of reduced coverage. The current abstract focuses exclusively on the accuracy improvements of the proposed method, creating a potentially misleading impression of its overall superiority. A balanced scientific report must present both the strengths and weaknesses of a new method. By omitting the substantial drop in data coverage, the authors fail to provide the reader with a complete picture of the algorithm's performance, which is essential for a fair and comprehensive initial assessment of the work.
  6. The manuscript fails to present individual-level results, obscuring the method's robustness. All presented results are aggregated at the group level. Given the extremely low average coverage for heart rate, it is highly probable that there is significant inter-patient variability, with the algorithm potentially failing completely for a subset of patients. Reporting only group-level statistics can mask poor performance or lack of robustness in individual cases. Providing individual-level data (e.g., in supplementary materials) is crucial for assessing how reliably the method performs across a heterogeneous patient population.
  7. The presentation of Bland-Altman plots could be improved for better clarity and quantitative assessment. The 95% limits of agreement in Figures 4 and 7 are depicted with dashed lines, which are difficult to distinguish amidst dense data points. The primary purpose of a Bland-Altman plot is to visually and quantitatively assess agreement. A clearer visual representation (e.g., using a shaded area) and the inclusion of a key quantitative metric—the percentage of outliers beyond the limits of agreement—would significantly enhance the interpretability and scientific value of these critical figures.
  8. The comparison with existing literature in Section 4.1 is not sufficiently rigorous. The authors claim their method is "among the best-performing contactless methods," but the comparison is made against studies involving different technologies (e.g., 60 GHz radar, PPG), populations (healthy vs. patients), and environments (lab vs. clinic). Such a direct, unqualified comparison can be misleading. A more scientifically sound approach would be to contextualize the results more precisely, highlighting the achievement within the specific constraints of the study (i.e., for this particular radar type, patient group, and setting). A structured comparison table would further aid in this nuanced interpretation.

Reviewer 2 Report

Comments and Suggestions for Authors

The authors present a well-structured study that complements previous work conducted by the same research team. In this study, an expanded database was used, including a significant number of patients with a history of acute ischemic stroke, and the performance of “multiple range bin selection” is compared with that of “single bin selection.” 

I believe that several aspects could improve the presentation of the study, as follows:

  1. Although the patient group corresponds to post-stroke individuals, the results do not relate any particular characteristics of this population. Therefore, referencing this population in the article title suggests that features specific to these patients—such as heart rate variability or distinctive ventilatory patterns—will be identified. In my view, the manuscript should either provide contributions in this regard or remove the specific reference to this patient group from the title.
  2. Section 2.9 describes a range of respiratory frequencies to be detected; it is advisable to discuss the limitation regarding the detection of apneas in such cases.
  3. Section 2.10 provides information about the standard PSG; however, no reference video is being recorded, and this should be specified.
  4. Section 2.10 details an incidence with a measurement script; however, I consider that this paragraph does not contribute to the study, particularly since it does not lead to the removal of any recordings.
  5. Section 2.10 reports the epoch-selection strategy of 60-s windows with 55-s overlap. This approach may affect apnea detection or heart rate variability analysis. Given its relevance in monitoring techniques, this issue should be discussed in the manuscript.

Round 2

Reviewer 1 Report

Comments and Suggestions for Authors

I have no further commnets

Reviewer 2 Report

Comments and Suggestions for Authors

The authors adequately addressed each of the concerns. I also consider their justification and the revisions made to be appropriate, as well as the inclusion of the supplementary material. In light of the above, I consider the publication of the article to be appropriate after the revisions made.